# Discovery of a novel cardiac-specific myosin modulator using artificial intelligence-based virtual screening

Priyanka Parijat[1], Seetharamaiah Attili[1], Zoe Hoare [2], Michael Shattock [2], Victor Kenyon [3] & Thomas Kampourakis [1] ✉

Direct modulation of cardiac myosin function has emerged as a therapeutic target for both heart disease and heart failure. However, the development of myosin-based therapeutics has been hampered by the lack of targeted in vitro screening assays. In this study we use Artificial Intelligence-based virtual high throughput screening (vHTS) to identify novel small molecule effectors of human β-cardiac myosin. We test the top scoring compounds from vHTS in biochemical counter-screens and identify a novel chemical scaffold called 'F10' as a cardiac-specific low-micromolar myosin inhibitor. Biochemical and bio-physical characterization in both isolated proteins and muscle fibers show that F10 stabilizes both the biochemical (i.e. super-relaxed state) and structural (i.e. interacting heads motif) OFF state of cardiac myosin, and reduces force and left ventricular pressure development in isolated myofilaments and Langendorff-perfused hearts, respectively. F10 is a tunable scaffold for the further development of a novel class of myosin modulators.

The contractile myofilaments drive and regulate the cardiac contraction-relaxation cycle. $Ca^{2+}$-activation of the actin-containing thin filaments leads to exposure of myosin binding sites on actin, which subsequently allows myosin head or motor domains from the neighboring thick filaments to strongly attach to actin and undergo the power-stroke fueled by the hydrolysis of ATP. The power-stroke leads to either nm-scale displacement of the thin filaments towards the centre of the sarcomere or the production of pN-scale forces, leading either to muscle shortening or force development on the cellular and organ level[1] (Fig. 1a). Conversely, $Ca^{2+}$-dissociation from the thin filaments triggers the detachment of myosin heads from actin and the onset of mechanical relaxation.

However, in addition to the classical $Ca^{2+}$-dependent thin filament regulatory pathway described above, activation of the thick filaments themselves has emerged as a second regulatory step that controls myofilament contractile function[2,3]. Similar to the thin filaments, thick filaments are believed to exist in both a diastolic OFF and systolic ON state, and the rate of transition between those states are likely rate-limiting for force development and mechanical relaxation[4].

The thick filament OFF state is structurally characterized by myosin heads sequestered onto the surface of the thick filaments in quasi-helical tracks, which is stabilized by both intra-molecular interactions between the two myosin heads of the dimeric myosin molecule and interactions between myosin heads and their coiled-coil tail domains, and intermolecular interactions between myosin heads on adjacent crowns[5,6]. Dimeric myosin molecules fold into an asymmetric conformation called the interacting-heads motif (IHM), with the so-called 'free head' sitting on top of the 'blocked head', obstructing its actin binding site and reducing its ATP hydrolysis rate[7–9] (Fig. 1a). Moreover, various intermolecular interactions with thick filament accessory proteins such as titin and cardiac myosin binding protein-C (cMyBP-C) have been shown to further stabilize the IHM and thick filament OFF state, and thereby regulate the number of myosin heads available for contraction[6,10]. For example, phosphorylation of cMyBP-C has been shown to increase the rate of ventricular force

[1]Randall Centre for Cell and Molecular Biophysics; and British Heart Foundation Centre of Research Excellence, King's College London, London SE1 1UL, United Kingdom. [2]School of Cardiovascular and Metabolic Medicine and Sciences; Rayne Institute and British Heart Foundation Centre of Research Excellence, King's College London, London SE5 9NU, United Kingdom. [3]Atomwise Inc., San Francisco, CA, USA. ✉e-mail: thomas.kampourakis@kcl.ac.uk

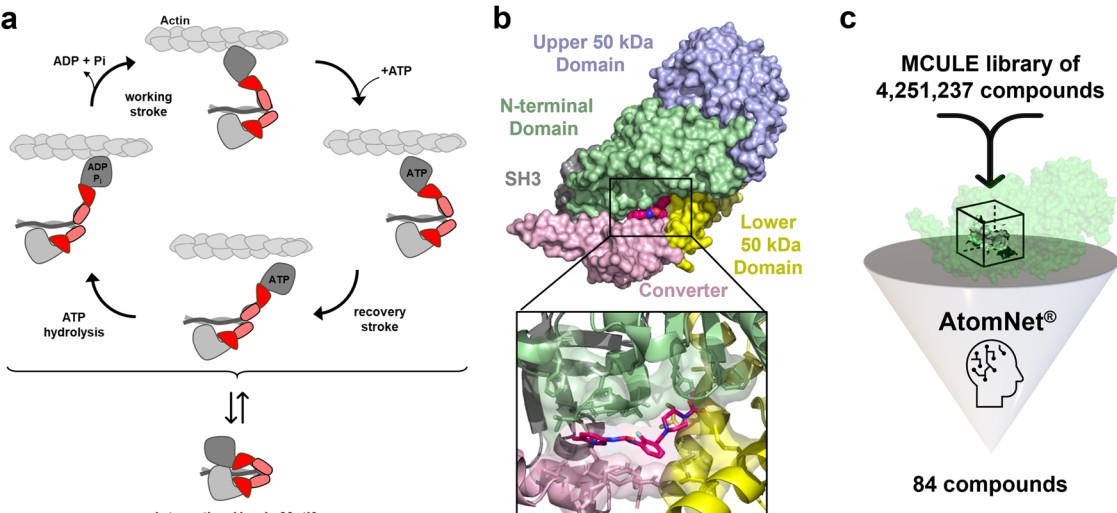

**Fig. 1 | Artificial Intelligence-based high throughput screen for cardiac myosin modulators. a** Cardiac myosin heads can either undergo their mechano-chemical cycle that produces force or muscle shortening (top), or can be sequestered into the functional OFF state (interacting heads motif, bottom). **b** Surface structure of human cardiac myosin S1 bound to Omecamtiv Mecarbil (OM, purple). Individual sub-domains of the myosin catalytic domain are labeled accordingly. **c** Artificial intelligence-based virtual screen for novel cardiac myosin modulators targeting the Omecamtiv Mecarbil-binding site. Several million compounds were virtually screened against the OM-binding site using AtomNet® technology. Top ranking 84 compounds were chosen for further investigation.

development during isovolumetric contraction likely by controlling the number of myosin heads available contraction and their associated cycling kinetics[11–13].

In addition to the structural OFF state, myosin heads can also adopt a functional or biochemical OFF state termed the 'super-relaxed state' (SRX), characterized by a very low intrinsic ATPase turn-over that is 10-100 lower compared to the rate observed for isolated myosin heads[14,15], potentially acting as an energy saving mechanism in the heart. However, the mechanistic link between the structural and functional OFF states of cardiac myosin remains to be established and recent studies suggested that interventions can modify one without affecting the other[16].

The functional significance of myosin filament-based regulation for the normal performance of the heart is further underlined by the fact that about 50% of patients suffering from inheritable Hypertrophic Cardiomyopathy (HCM) carry mutations in the genes encoding for either cardiac myosin (MYH7) or cardiac myosin binding protein-C (MYBPC3)[17]. In good agreement, ablation of either cMyBP-C or its phosphorylation has been shown to lead to heart disease and heart failure in transgenic animal models[18,19], likely by dysregulation of the thick filaments.

It is therefore not surprising that direct modulation of cardiac myosin function has emerged as a promising new route for the development of novel classes of therapeutic interventions for both heart disease and heart failure[20–22]. In contrast to classical pharmacological heart failure therapies based on neurohumoral modulation, directly changing myosin filament function is thought to be a more efficient therapy, targeting the underlying etiologies rather than focusing on symptomatic relief. Moreover, direct myosin modulation potentially circumvents some of the side effects associated with current therapies such as increased energy wastage and oxygen consumption of the myocardium, as well as arrythmias and bradycardias/tachycardias[23,24]. The broad spectrum of disease-related mutations in sarcomeric proteins and associated disease mechanisms further suggests that myosin-targeted therapies can be developed towards a personalized medicine approach.

In agreement, cardiac myosin modulators with both activating and inhibiting effects have been developed for the treatment of both systolic and diastolic heart failure[20,21]. Omecamtiv Mercarbil (OM) was developed as a cardiac myosin activator for the treatment of systolic heart failure. OM has been shown to increase actomyosin attachment rate by stabilizing the functional ON state of the myosin motors and thick filament, which in turn increases the calcium sensitivity of the myofilaments by cooperative thin filament activation[25,26]. However, OM treatment has been associated with increased resting myocardial oxygen consumption and energy wastage[27]. Recently, the FDA rejected the approval of OM for heart failure with reduced ejection fraction (HFrEF). In contrast, Mavacamten (Mava; now called 'Camzyos') was approved by the FDA as the first cardiac myosin inhibitor for the treatment of obstructive Hypertrophic Cardiomyopathy subtype. Mava has been shown to reduce cardiac contractility by stabilizing both the biochemical and structural OFF state of cardiac myosin[28], and treatment can prevent the onset of HCM disease in animal models[21].

In the current study we use Artificial Intelligence-based virtual screening to identify new compounds that can modulate cardiac myosin function. We screen a large virtual library of millions of compounds against the determined OM-binding site on human cardiac myosin and identify a novel chemical scaffold (called 'F10') in a biochemical counter-screen as a cardiac-specific myosin modulator. Surprisingly, F10 acts as a cardiac-myosin specific inhibitor that, similar to Mava, stabilizes both the biochemical and structural OFF state of the cardiac myosin motor domains, and reduces force production and calcium sensitivity of cardiac muscle in vitro. Moreover, F10 acts as a negative inotrope in isolated Langendroff-perfused rat hearts. Using computational docking we show that F10's hydrophobic head group binds deep within the proposed OM binding site, whereas its hydrophilic tail can form interactions with various sub-domains of the myosin motor domain including the converter and N-terminal domain. We propose that F10 is a new tuneable scaffold for the development of novel myosin modulators.

## Results

### Artificial intelligence-based virtual screen for cardiac myosin effectors

We searched the PDB archive for suitable starting structures for the virtual high throughput screen (vHTS) against the motor domain of human β-cardiac myosin (MYH7). Holo-structures of protein-ligand complexes are preferred targets over protein apo structures in vHTS

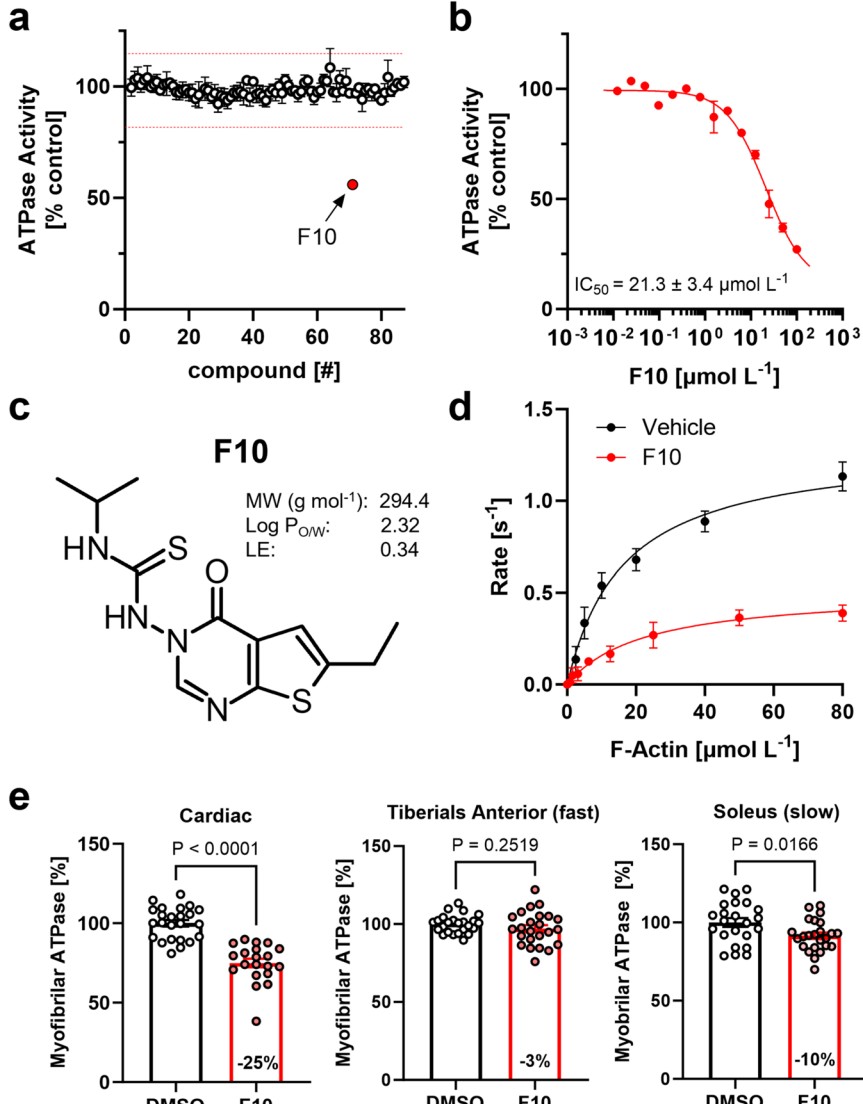

**Fig. 2 | Identification of F10 as a cardiac-specific allosteric myosin inhibitor.**
**a** Single dose acto-myosin S1 ATPase activity screen of the 84 top ranking compounds from AI-based virtual screen. Hit thresholds are shown as red dashed lines. Means ± s.e.m. for $n = 4$ independent repeats, except for compounds #46 and #49 with $n = 3$ independent repeats. **b** Dose-response analysis for hit compound F10 on acto-myosin S1 ATPase activity. Means ± s.e.m., $n = 6$ independent repeats. **c** Chemical structure and physico-chemical properties of F10 (MW – molecular weight; log $P_{O/W}$ – partition coefficient octanol/water; LE – ligand efficiency). **d** F-Actin concentration-dependent bovine cardiac myosin S1 ATPase activity in the absence (black, vehicle control) and in the presence of 100 µmol/L F10 (red). Means ± s.e.m., $n = 3$ independent repeats for control and $n = 9$ independent repeats in the presence of F10. **e** ATPase activity of myofibrils isolated from cardiac, and fast and slow skeletal muscle in the absence (black) and in the presence of 20 µmol L$^{-1}$ F10. Means ± s.e.m. for $n = 20$-23 experiments from $n = 3$ independent myofibril preparations. Statistical significance between control and drug treatment group was assessed with a two-tailed, unpaired student's t-test for parametric data or Mann-Whitney test for non-parametric data sets. Source data are provided as a Source Data file.

because of a better-defined binding geometry which increases the probability of identifying biologically active molecules[29,30]. We therefore chose the nucleotide free-structure of human β-cardiac myosin bound to Omecamtiv Mecarbil (OM; PDB entry 4PA0) as the starting template[31] (Fig. 1b). The OM-binding site is solvent accessible and at sufficient distance to both the ATP- and actin-binding sites, making it an ideal target for the AI-based virtual screen for an allosteric effector. Moreover, the binding site shows a variety of probable hydrogen bond donors and acceptors, as well as hydrophobic characteristics, increasing the likelihood of identifying a specific molecular binder.

To identify novel myosin modulators, a virtual library of approximately four million compounds was evaluated, classified, and ranked according to probability of binding to the Omecamtiv Mercarbil binding site on the human β-cardiac myosin motor domain. The top ranked compounds were filtered to remove molecules with undesirable properties (e.g., fragments, fused ring systems, logP > 5) and selected for purchase. Of the top 200 ranked molecules, 84 compounds and two negative DMSO controls were shipped for empirical validation in a blinded manner, with only the well location as an identifier (Fig. 1c).

**Identification of F10 as a novel cardiac-specific myosin inhibitor**
86 blinded samples, including the 84 highest ranking compounds from the AI-based virtual screen and two DMSO controls, were tested in a single dose screening assay measuring the ATPase activity of isolated bovine β-cardiac myosin S1 in the presence of 10 µmol L$^{-1}$ F-actin (Fig. 2a). Sequence analysis showed that the OM binding site is highly conserved between human and bovine β-cardiac myosin (Fig. S1), suggesting that the bovine isoform is a suitable replacement for the human protein in the screening assays.

We identified one compound, subsequently referred to as F10, that reproducibly reduced the F-actin stimulated ATPase activity of bovine β-cardiac myosin S1 by 44 ± 2% (mean ± s.e.m., $n = 4$ for independent repeats) at a concentration of 10 μmol L$^{-1}$. The positive hit was further confirmed by dose-response analysis, showing that F10 maximally decreased the actomyosin ATPase activity by 89 ± 5% with an IC$_{50}$ of 21 ± 3 μmol L$^{-1}$ (means ± s.e.m., $n = 6$) (Fig. 2b). The chemical structure of F10 is shown in Fig. 2c.

We searched both the PubChem and ChEMBL databases for any known biological activity of F10 without, however, any result[32,33]. We therefore compared F10 to structures of known myosin-directed small molecule effectors using a multi-dimensional scaling approach in ChemMineTools[34] (Fig. S2). Intriguingly, F10 did not cluster with any of the published structures, suggesting that it might represent a novel chemical scaffold. The highest similarity was observed for Mavacamten with a Tanimoto coefficient of about 0.22. Moreover, F10 has excellent predicted ADMET properties and drug likeness[35] (Fig. S3).

To further test for the effects of F10 on the steady-state ATPase activity, we measured the dependence of the ATPase activity of β-cardiac myosin S1 on the F-actin concentration in the absence and in the presence of the F10 compound (Fig. 2d). F10 significantly reduced the maximal rate of ATP hydrolysis (k$_{cat}$) from about 1.3 s$^{-1}$ head$^{-1}$ to about 0.5 s$^{-1}$ head$^{-1}$, without, however, affecting the apparent dissociation equilibrium constant for F-actin binding (K$_{app}$ of about 20 μmol L$^{-1}$ in the absence and in the presence of F10). This suggests that although F10 reduced the maximum rate of ATP hydrolysis, it did not affect the affinity of myosin S1 for F-actin per se.

Next we tested the myosin isoform specificity of F10 by measuring its effect on the ATPase activity of myofibrils isolated from ventricle, and fast (*Tiberialis Anterior*) and slow skeletal muscles (*Soleus*) isolated from marmoset (*Callithrix jacchus*). As shown in Fig. 2e, 20 μmol L$^{-1}$ F10 significantly reduced the ATPase activity of fully Ca$^{2+}$-activated cardiac myofibrils by about 25% but had no effect on the ATPase of myofibrils isolated from fast skeletal muscle. In contrast, F10 had an intermediate effect on myofibrils isolated from soleus muscle, corresponding to a decrease in ATPase activity by about 10%. This is in very good agreement with the distribution of the myosin heavy chain isoforms in the different striated muscle types[36], suggesting that F10 is a MYH7/6-specific inhibitor.

## F10 slows nucleotide release from cardiac myosin motors

As a step towards elucidating the molecular mechanism behind F10's inhibitory effect on the actin-activated ATPase activity of cardiac myosin, we performed single nucleotide turnover experiments using the mant-ATP chase assay[14] (Fig. 3). We used systems of increasing complexity ranging from the isolated cardiac myosin motor domains (Fig. 3a) to synthetic thick filaments formed from purified full-length bovine β-cardiac myosin (Fig. 3b) and bovine cardiac myofibrils as substrates in the chase assays (Fig. 3c). In contrast to the isolated myosin motors, synthetic thick filaments contain two myosin head domains per molecule connected via their coiled-coil tail. Both head-head and head-tail interactions have been shown to be a requirement for the myosin heads to adopt the folded OFF conformation, also called the interacting heads motif (IHM), as shown by electron microscopy reconstructions of isolated native thick filaments[5]. Moreover, myofibrils contain intact thin and thick filaments organized in the native myofilament lattice, and thick filament accessory proteins like cardiac myosin binding protein-C (cMyBP-C) and titin have been shown to be able to control the regulatory state of the myosin motors[37].

Chase of mant-ATP loaded cardiac myosin S1 with ATP leads to a decrease in the mant fluorescence intensity which is best described by a bi-exponential decay function with an initial fast phase followed by a second slower phase, corresponding to the so-called disordered (DRX) and super-relaxed state (SRX) of myosin, respectively. In the absence of compounds, the majority of myosin motor domains (79 ± 2%, mean

± s.e.m., $n = 5$) are found in the DRX state with an ATP hydrolysis rate of 0.05 ± 0.01 s$^{-1}$ (Fig. 3a, Supplementary Table 1). The remaining ~20% of myosin motors occupy the SRX state with an about ten-fold lower ATP turn-over rate (0.006 ± 0.001 s$^{-1}$), in good agreement with previously published results[38].

Mavacamten (Mava) has previously been shown to stabilize the SRX state of isolated cardiac myosin and we used the compound as a positive control in our assays[28]. Addition of Mava increased the amplitude of the slow phase to 71 ± 6% (mean s.e.m., $n = 6$) and decreased its ATP hydrolysis rate by a factor of about two (0.0035 ± 0.0002 s$^{-1}$; Supplementary Table 1). Mava had no effect on the rate of the fast DRX phase. Similar results were obtained in the presence of F10 with a slow phase amplitude of 70 ± 5% (mean s.e.m., $n = 6$). However, F10 had a significantly stronger effect on the rate of the slow phase with a more than four-fold reduction compared to the control in the absence of compounds (0.0014 ± 0.0002 s$^{-1}$). This suggests that Mava and F10 might stabilize the SRX state of cardiac myosin S1 via distinct mechanisms.

Similar results were obtained using synthetic thick filaments formed from purified full-length bovine β-cardiac myosin (Fig. 3b). Mava and F10 increased the amplitude of the slow SRX phase from about 23% under control conditions to about 71% and 63%, respectively. As before, both Mava and F10 had no effect on the rate of the fast DRX phase but decreased the rate of the slow phase, which however did not reach statistical significance (Supplementary Table 1).

Mava and F10 also increased the amplitude of the slow phase in isolated bovine cardiac myofibrils (Fig. 3c and Supplementary Table 1). However, the effect on the slow phase amplitude was significantly weaker for Mava compared to F10, with an increase from about 25% under control conditions to about 47% and 68%, respectively. As before, Mava and F10 reduced the rate of the slow phase by about a factor of two but had no effect on the rate of the fast phase.

## Functional characterization of F10 in demembranated cardiac muscle fibers

The results presented above show that F10 is an allosteric inhibitor of isolated cardiac myosin that stabilizes its biochemical OFF or SRX state. Next, we tested for the functional effects of F10 in demembranated rat ventricular trabeculae with intact thick and thin filaments organized into the native myofilament lattice. Similar to bovine β-cardiac myosin, the OM-binding site is highly conserved between human β- and rat α-cardiac myosin (Fig. S1).

Although 20 μmol L$^{-1}$ F10, a concentration close to the IC$_{50}$ for the ATPase activity of the isolated myosin motor domain, had no effect on the passive tension of demembranated rat trabeculae during relaxing conditions at pCa 9 (pCa = -log$_{10}$[Ca$^{2+}$]), it decreased the maximal active isometric tension at full Ca$^{2+}$-activation (pCa 4.5) by more than 75% from 69.6 ± 2.6 mN mm$^{-2}$ to 15.6 ± 1.8 mN mm$^{-2}$ (means ± s.e.m., $n = 5$ trabeculae) (Fig. 4a. Supplementary Table 2). In a subset of trabeculae, we replaced the endogenous myosin regulatory light chain (RLC) with a recombinant cardiac RLC crosslinked to a bifunctional sulforhodamine (BSR) along its E-helix (BSR-cRLC-E)[39,40]. This allowed us to determine the orientation of the RLC-region of myosin in demembranated trabeculae using polarized fluorescence (Fig. 4b; Supplementary Table 2). The order parameter <P$_2$> describes the orientation of the probe fluorescence dipole with respect to the filament axis[41] and decreases for BSR-cRLC-E going from the relaxed (pCa 9) to the fully Ca$^{2+}$-activated state (pCa 4.5), suggesting that the myosin heads become more perpendicular with respect to the filament axis[39,40]. In contrast, F10 increased <P$_2$> for the E-helix probe, under both relaxing and activating conditions, and abolished the orientation change in the myosin heads associated with Ca$^{2+}$-activation of the myofilaments. This suggests that F10 stabilizes the parallel orientation of the myosin motors as seen in electron microscopy reconstructions of isolated thick filaments in the OFF state[5]. As a step towards

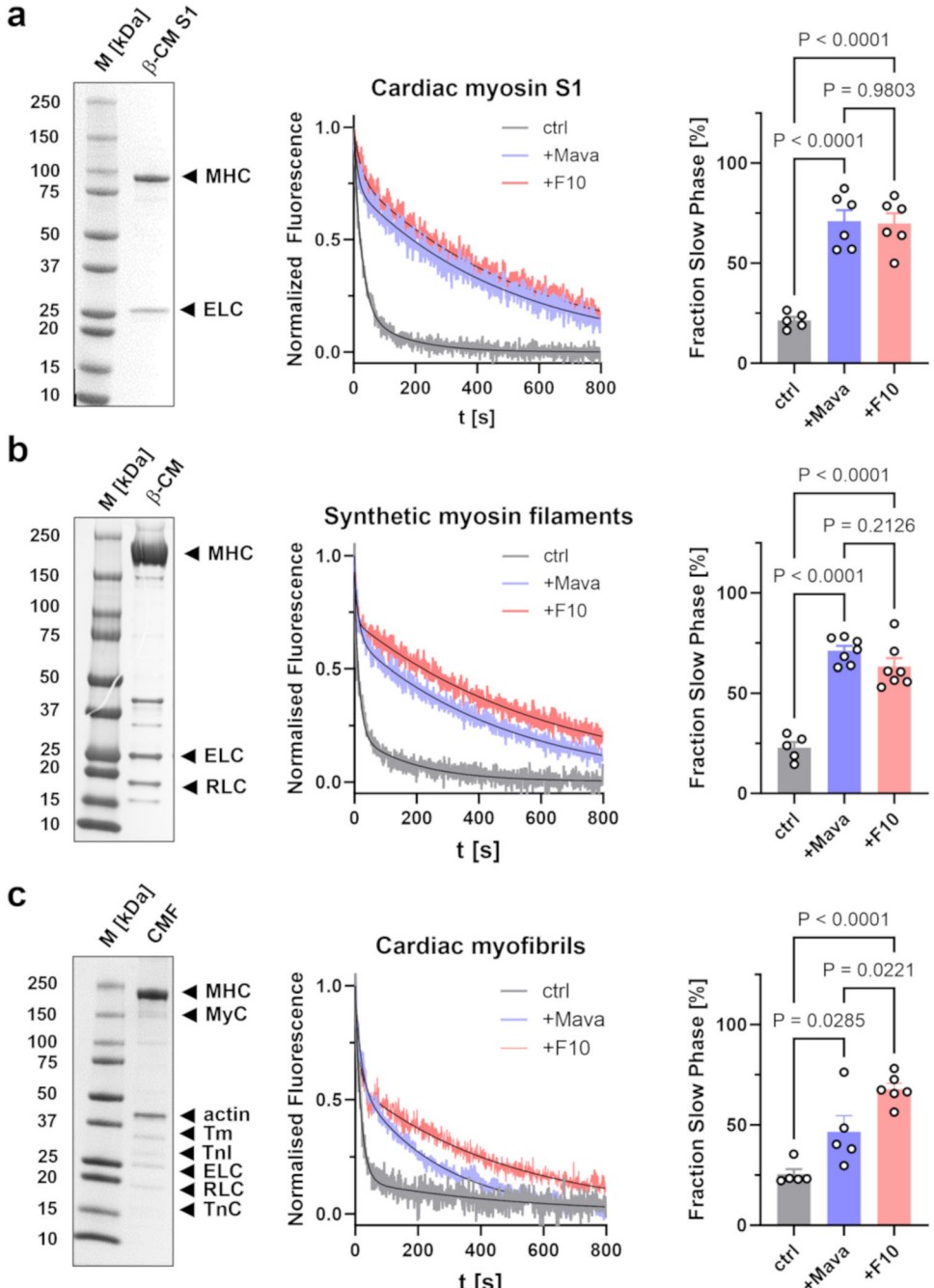

**Fig. 3 | F10 slows nucleotide release from bovine β-cardiac myosin. a** Left: SDS-PAGE of isolated bovine cardiac myosin S1 with essential light chain (ELC) and myosin heavy chain (MHC) labeled accordingly. Middle: Representative traces of mant-ATP chase experiments in the presence of vehicle control (gray), 2 µmol L$^{-1}$ Mavacamten (purple) or 20 µmol L$^{-1}$ F10 (red). Bi-exponential fits to data are shown as black continuous lines. Right: Fraction of slow phase nucleotide release for the three conditions. Means ± s.e.m., $n = 5$ independent repeats for control, and $n = 6$ independent repeats in the presence of F10 or Mavacamten. **b** Left: SDS-PAGE of isolated bovine cardiac myosin with regulatory light chain (RLC), essential light chain (ELC) and myosin heavy chain (MHC) labeled accordingly. Middle: Representative traces of mant-ATP chase experiments in the presence of vehicle control (gray), Mavacamten (purple) and F10 (red). Bi-exponential fits to data are shown as black continuous lines. Right: Fraction of slow phase nucleotide release. Means ± s.e.m., $n = 5$ independent repeats for control, and $n = 7$ independent repeats in the presence of F10 or Mavacamten. **c** Left: SDS-PAGE of isolated bovine cardiac myofibrils with main proteins labeled accordingly (MHC – myosin heavy chain, MyC – myosin binding protein-C, Tm – tropomyosin, TnI – troponin I, ELC – essential light chain, RLC – regulatory light chain, TnC – troponin C). Middle: Representative traces of mant-ATP chase experiments in the presence of vehicle control (gray), 2 µmol L$^{-1}$ Mavacamten (purple) or 20 µmol L$^{-1}$ F10 (red). Bi-exponential fits to data are shown as black continuous lines. Right: Fraction of slow phase nucleotide release for the three conditions. Means ± s.e.m., $n = 5$ independent repeats for control, $n = 6$ independent repeats in the presence of F10 and $n = 6$ independent repeats in the presence of Mavacamten. Statistical significance of differences between groups were assessed with a one-way ANOVA followed by Tukey's post-hoc test. Source data are provided as a Source Data file.

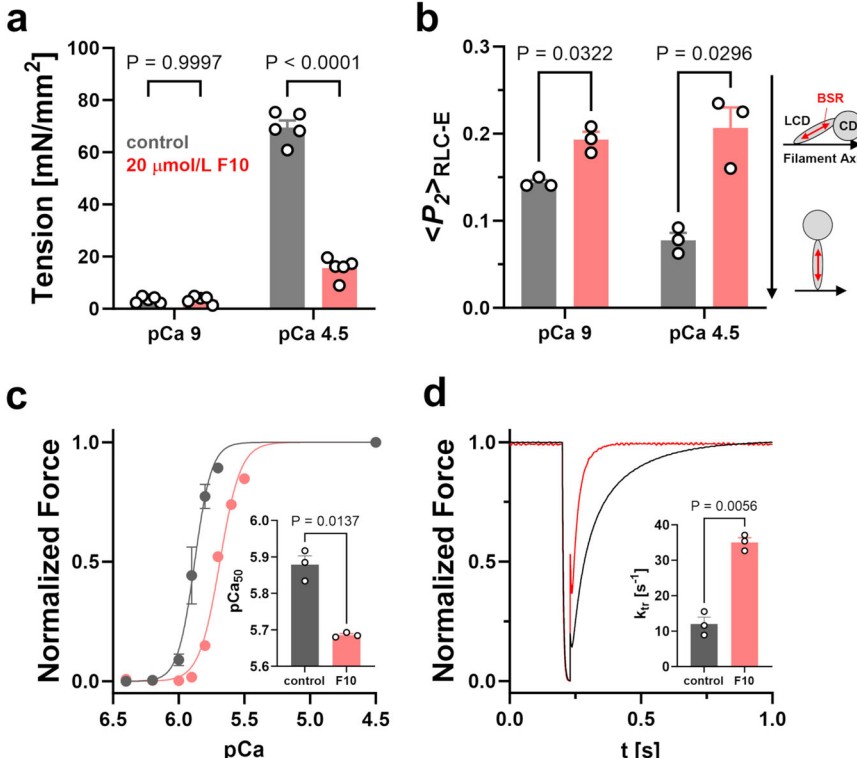

**Fig. 4 | Effect of F10 on cardiac muscle mechanics and thick filament structure.**
**a** Isometric force of demembranated rat ventricular trabeculae during relaxing conditions (pCa 9) and full $Ca^{2+}$ activation (pCa 4.5) in the absence (gray) and in the presence of 20 µmol $L^{-1}$ F10. Means ± s.e.m., $n = 5$ trabeculae. **b** Order parameter $<P_2>$ determined from polarized fluorescence of BSR-cRLC-E exchanged demembranated rat trabeculae during relaxing conditions (pCa 9) and full $Ca^{2+}$ activation (pCa 4.5) in the absence (gray) and in the presence of 20 µmol $L^{-1}$ F10. Means ± s.e.m., $n = 3$ trabeculae. Pictograms indicate the orientation of the BSR probe (red double arrow) and myosin heads (CD, catalytic domain; LCD, light chain domain,

BSR – Bifunctional sulforhodamine) with respect to the thick filament axis (black arrow). **c** Normalized force-pCa relation of demembranated trabeculae in the absence (gray) and in the presence of F10 (red). Inset: Extracted $pCa_{50}$ values before and after F10 treatment. Means ± s.e.m., $n = 3$ trabeculae. **d** Example traces of force re-development of demembranated trabeculae following a slack-stretch protocol in the absence (gray) and in the presence of F10 (red). Inset: Rates of force re-development ($k_{tr}$). Means ± s.e.m., $n = 3$ trabeculae. Statistical significance of differences between values before and after drug treatment were assessed with a paired, two-tailed student's t-test. Source data are provided as a Source Data file.

elucidating the molecular mechanism behind F10's stabilizing effect on the myosin head OFF conformation, we measured the affinity of isolated bovine β-cardiac myosin S1 in the presence of various nucleotides and F10 for the myosin S2 tail domain using Microscale Thermophoresis (Fig. S4). Surprisingly, F10 had no effect on the myosin S1-S2 binding affinity with steady state dissociation constants $K_d$ of about 15 µmol $L^{-1}$ under all conditions tested.

Modulation of the thick filament regulatory state has been previously shown to change myofilament calcium sensitivity[25]. In good agreement, F10 treatment decreased the calcium sensitivity of force development of rat ventricular trabeculae as indicated by a change in the $pCa_{50}$ (pCa that produces half-maximal force) from 5.88 ± 0.02 to 5.69 ± 0.01 (means ± s.e.m., $n = 3$ trabeculae) (Fig. 4c; Supplementary Table 2). F10 decreased active isometric force at pCa 5.9 (corresponding to about 50% maximal activation) in a concentration-dependent manner, which can be described by a sigmoidal dose-response with an $IC_{50}$ of about 9 µmol $L^{-1}$ (Fig. S5). Moreover, drug treatment slightly reduced the steepness of the force-pCa relation as indicated by a decrease in the Hill coefficients ($n_H$) from about 7.6 under control conditions to about 5.6 in the presence of F10 (Supplementary Table 2).

We measured the effect of F10 on crossbridge kinetics using a slack-re-stretch protocol (Fig. 4d; Supplementary Table 2). In the absence of drug, force re-develops after the re-stretch with an exponential time-course and a rate constant ($k_{tr}$) of 12 ± 2 $s^{-1}$ (mean ± s.e.m, $n = 3$ trabeculae). Strikingly, F10 significantly increased $k_{tr}$ to 35 ± 1 $s^{-1}$, suggesting about three-times faster crossbridge kinetics.

## Functional effects of F10 in Langendorff-perfused rat hearts

Next, we tested for the functional effects of F10 in isolated Langendorff-perfused rat hearts. Hearts were initially stabilized in modified Krebs-Henseleit (KH) solution for about 25–30 min and subsequently perfused with KH solution containing 20 µmol $L^{-1}$ F10 for 5 mins to reach a new steady state. As shown in Fig. 5, perfusion with F10 rapidly decreased left ventricular systolic pressure (LVSP) by about 65% from about 170 mmHg to 60 mmHg within 2–3 mins without, however, affecting either heart rate (HR) or coronary perfusion (CP) (Table S3). Moreover, F10 perfusion significantly reduced both the maximum and minimum rate of pressure development ($dP/dt_{max}$ and $dP/dt_{min}$, respectively), suggesting a slowed time-course of pressure development (Fig. 5 and Supplementary Table 3). Strikingly, the effect of F10 is completely reversible and perfusion with KH solution in the absence of F10 restores haemodynamic parameters of the isolated heart preparation to baseline values within three minutes.

We compared the effects of F10 to the FDA-approved myosin inhibitor Mavacamten by perfusing isolated rat hearts with KH solution containing 1 µmol $L^{-1}$ Mavacamten (Fig. 5, right column), a concentration close to reported $IC_{50}$ for force inhibition in isolated cardiac muscle fibers[21]. Similar to F10, Mavacamten reduced LVSP, and $dP/dt_{max}$ and $dP/dt_{min}$, without affecting either heart rate or coronary flow. Interestingly, however, Mavacamten did not affect the left ventricular end-diastolic pressure (LVEDP), which was slightly increased in the presence F10. More strikingly, however, both the ON and OFF rate for the effect of Mavacamten on left ventricular systolic pressure (LVSP) were vastly slower compared to F10. The changes of LVSP in the

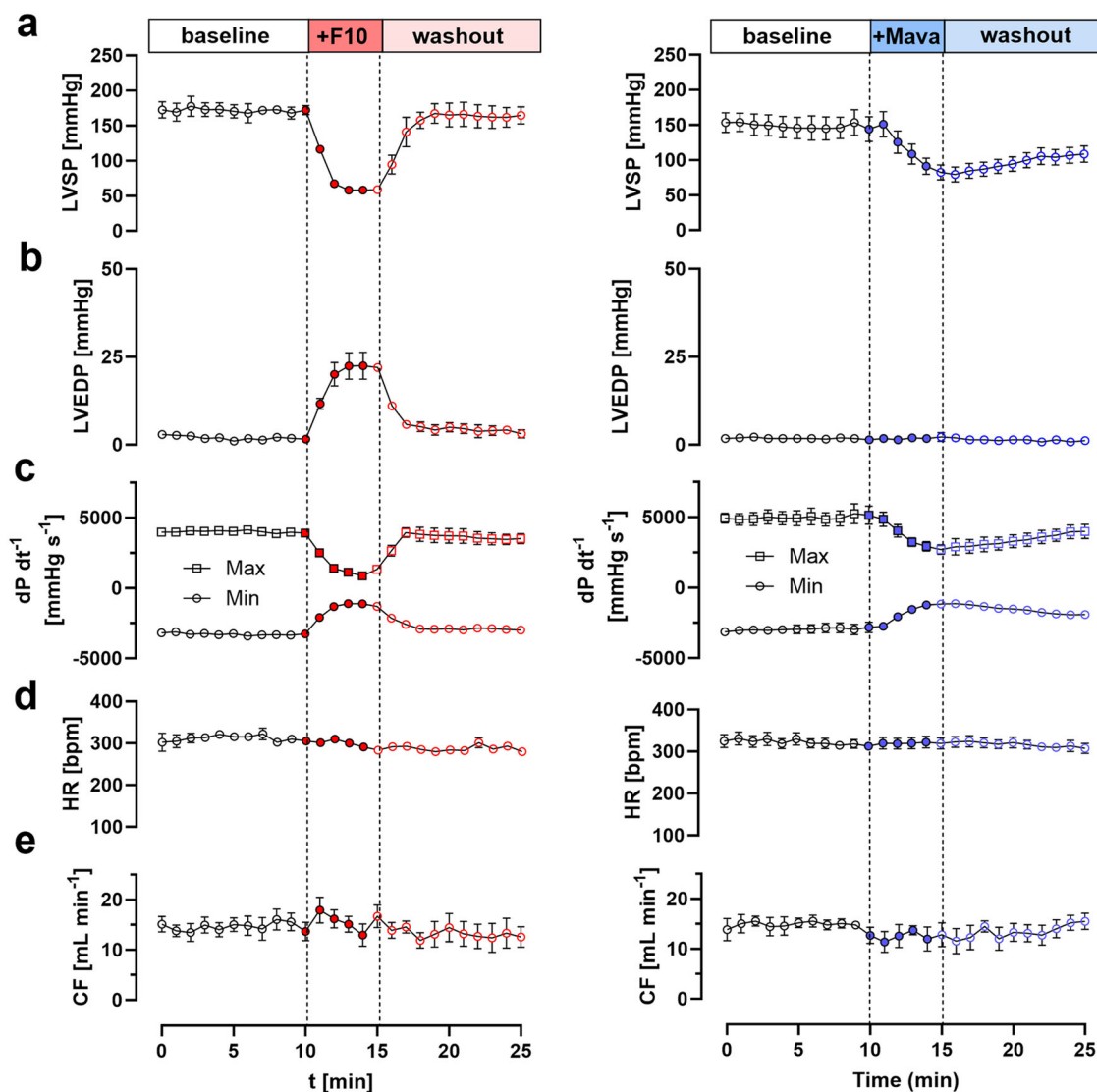

**Fig. 5 | Effect of F10 and Mavacamten on haemodynamic parameters of Langendorff-perfused rat hearts. a** Left ventricular systolic pressure (LVSP), **b** left ventricular end-diastolic pressure (LVEDP), **c** maximum and minimum rate of pressure development (dP dt$^{-1}_{max}$ and dP dt$^{-1}_{min}$, respectively), **d** heart rate (HR) and **e** coronary flow (CF) in the absence of compounds (open circles) and in the presence of either 20 μmol L$^{-1}$ F10 (left) or 1 μmol L$^{-1}$ Mavacamten (right), and after washout. Data represent means ± s.e.m. with $n = 4$ rat hearts for F10 and $n = 5$ rat hearts for Mavacamten. Source data are provided as a Source Data file.

presence of either F10 or Mavacamten were well described by a Boltzmann sigmoidal function (Fig. S6) and allowed us to determine the half-time for both ON and OFF rate. F10 decreased LVSP with a half-time of about 45 s, which is about three times faster compared to Mavacamten with a half-time of 120 s. Similarly, the Mavacamten OFF rate was about four-times slower compared to F10 ( ˜ 200 s vs ˜50 s for Mavacamten and F10, respectively).

Taken together, these results suggests that F10 exhibits a strong negative inotropic effect on heart muscle function with fast pharmacodynamics, which is in excellent agreement with the acto-myosin ATPase and demembranated trabeculae experiments described above.

### Structure-activity relationship analysis of F10 gives insights into its molecular mechanism of action

The results shown above suggest that F10 is a potent inhibitor of cardiac myosin that can stabilize both its biochemical (i.e. the SRX state) and structural OFF state (i.e. interacting heads motif), and reduces both isometric force and its calcium sensitivity in isolated muscle fibers and acts as a negative inotrope in Langendorff-perfused hearts. To better understand the molecular mechanism of action of F10 on

cardiac myosin motor domain function, we performed computational docking of F10 into the OM-binding site using AutoDock Vina[42]. The five highest scoring docking poses are shown superimposed onto the β-cardiac myosin S1 structure in Fig. 6a. The ethyl-group connected to the pyrimidine moiety of F10 binds deep within the hydrophobic pocket formed by the N-terminal (NTD) and lower 50 kDa domain (L50D) in all docking poses, suggesting a likely representation of the bound conformation. In contrast, the orientation of F10's thiourea-based tail is less well defined and varies significantly between the five docking poses, likely associated with the higher flexibility of this region of the molecule. This suggests that F10 might interact with the myosin motor in more than one conformation, each making contact with different residues on the NTD (Ala91), L50D (Glu496 and L497) and converter domain (Pro710 and Lys712), and that those interactions are responsible for its functional effect on the myosin motor.

We tested this idea by purchasing commercially available derivatives of F10 with different modifications to its thiourea moiety (Fig. 6b) and used these compounds in dose-dependent activity screens measuring the ATPase activity of bovine cardiac myofibrils (CMF) at suboptimal Ca$^{2+}$-activation, corresponding to about 70% maximal activation

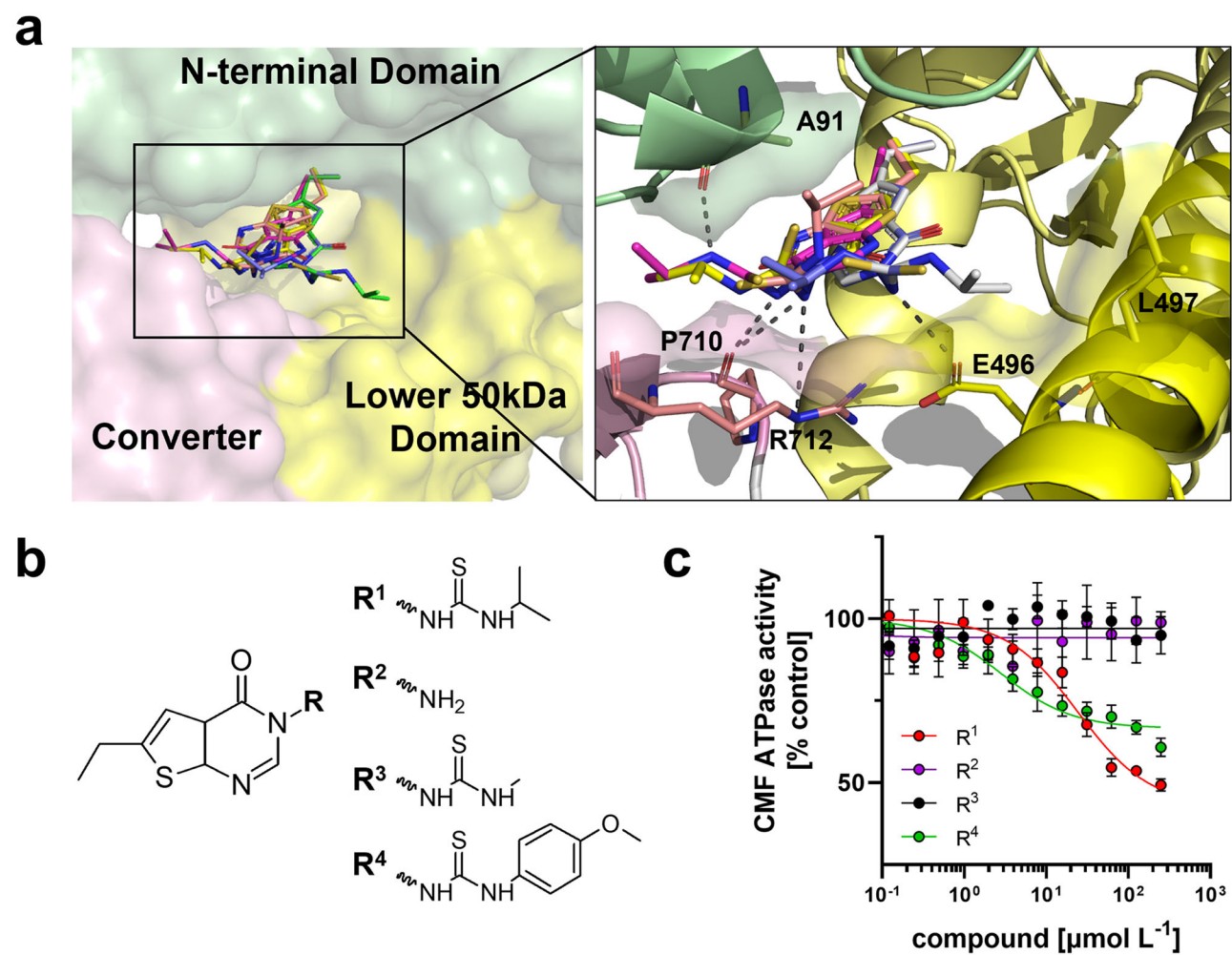

**Fig. 6 | Structure-activity relationship of F10. a** Left: Top five highest scoring docking poses of F10 (stick representation) in the OM-binding site of human cardiac myosin S1 created using AutoDock Vina. The N-terminal domain (NTD), converter and lower 50 kDa domain (L50D) are shown in green, pink and yellow, respectively. Right: Schematic showing the proposed main interactions of F10 within the myosin catalytic domain. Potential hydrogen bonds are indicted by black dashed lines. **b** Chemical structures of tested F10 derivatives. **c** Dose-response analysis for the effect of F10 derivatives on bovine cardiac myofibrillar ATPase activity. Means ± s.e.m., n = 7 independent experiments. Source data are provided as a Source Data file.

(Fig. 6c). The parent compound F10 inhibited the ATPase activity of bovine CMF with an IC$_{50}$ of about 24 μmol L$^{-1}$, in very good agreement with the results for the acto-myosin ATPase described above, and a maximal inhibition of about 60%. A shorter variant without the thiourea moiety had no activity in the concentration range tested. Similarly, replacing the terminal isopropyl with a methyl group resulted in complete loss of activity, suggesting that a larger hydrophobic terminal group is required for F10's inhibitory activity. In good agreement, replacing the isopropyl group with a larger hydrophobic methoxyphenyl moiety decreased the IC$_{50}$ for CMF ATPase activity by about a factor of ten (IC$_{50}$ of about 3 μmol/L), albeit with a lower maximal inhibition of about 40%.

Taken together, these results suggest that the OM binding site in cardiac myosin can be targeted for the development of myosin-based small molecule effectors, and that F10 is a tuneable scaffold for the development of a new class of myosin modulators.

## Discussion

The results presented above show that Artificial Intelligence-based virtual screening for cardiac myosin modulators is a viable alternative to traditional phenotypical or target-based screening campaigns[43,44]. Not only are virtual screens more time- and cost-efficient but also have access to a larger chemical space, increasing the probability of finding biologically-active molecules[45]. Although currently only one out of 84 top-scoring compounds identified from a library of about 4.25 million showed a desirable biological activity, the availability of ultra-large virtual libraries with several billions of compounds suggests that potentially hundreds of new chemical scaffolds can be discovered in future virtual screening campaigns[46,47]. Moreover, vHTS campaign are not limited to known binding sites and tools for the identification of functionally important binding pockets are readily available[48].

In the present study, we identified F10 as a cardiac-specific myosin modulator using an AI-based virtual screen against the Omecamtiv Mecarbil-binding pocket in the nucleotide-free human β-cardiac myosin motor domain (Fig. 1). The identified compound exhibits IC$_{50}$ for both myosin, acto-myosin and myofibrillar ATPase in the low micromolar range, comparable to the efficacy and potency of myosin-targeted small molecule effectors identified in traditional wet lab-based primary screens[43,44]. Moreover, F10 shows excellent drug-likeness and predicted ADMET properties (Fig. S3).

Perhaps the most surprising aspect of the current results, however, is that although OM is generally considered a cardiac myosin activator[20], F10 acts as an allosteric inhibitor of the cardiac myosin and actomyosin ATPase, and reduces force production and calcium sensitivity in isolated cardiac myofilaments (Figs. 3 and 4). Moreover, F10 reduces left ventricular pressure development and slows its timecourse in isolated

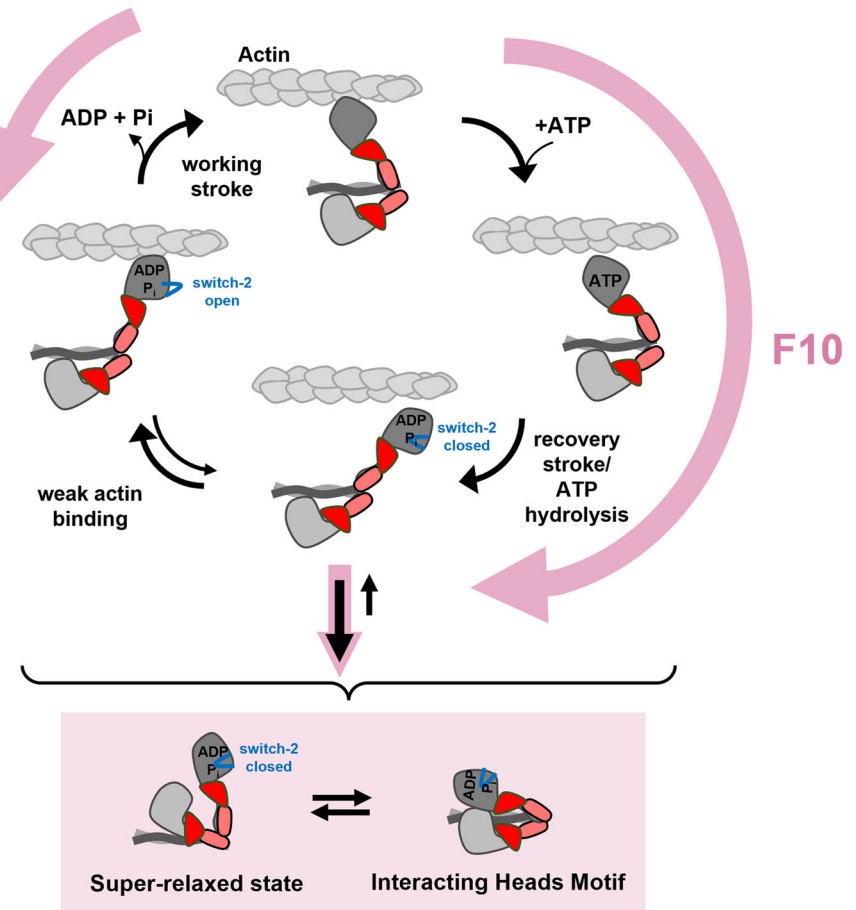

**Fig. 7 | Proposed mechanism for the effect of F10 on cardiac myosin function.** F10 accelerates myosin head detachment from actin and prevents the transition from weakly- to strongly-bound crossbridges, and stabilizes its structural (interacting heads motif) and functional OFF state (super-relaxed state).

Langendorff-perfused rat hearts (Fig. 5), suggesting a strong negative inotropic effect on heart muscle function. Strikingly, F10 exhibits a very high ON and OFF rates in Langendorff-perfused hearts, which are several folds faster than the FDA-approved myosin inhibitor Mavacamten, suggesting potentially faster clinical action and washout.

We show that F10 slows the nucleotide release rate from isolated cardiac myosin motors, synthetic thick filaments and relaxed myofibrils, strongly suggesting that it stabilizes the so-called super-relaxed state of cardiac myosin associated with the switch-2 closed conformation of the myosin catalytic domain in the ADP.$P_i$ conformation[14,15] (Fig. 3). F10 therefore likely increases the fraction of myosin motors in the ADP.$P_i$ low force/weak binding state and prevents their transition into the strong-bound force-generating states (Fig. 7), consistent with the unaltered apparent affinity of the myosin motors for actin in the presence of F10 but strong reduction in ATPase activity and force generation (Fig. 2d). In agreement, our fluorescence polarization data show that F10 stabilizes the parallel orientation of the myosin motors suggesting that it favors the formation of the IHM state of the double headed myosin molecule and the thick filament OFF state (Fig. 4b). The increased fraction of myosin motors in the SRX and IHM state leads to depressed force generation and pressure development in isolated cardiac muscle fibers and Langendorff-perfused hearts, respectively.

Although the mechanistic link between the super-relaxed state and the IHM motif has not been established, previous studies suggested that structural changes in the catalytic domain of the myosin heads associated with the SRX might promote the formation of the structural OFF state[14,28,49]. Recent cryo-electron microscopy reconstructions of mammalian cardiac thick filaments showed a multitude of inter- and intra-molecular interactions that stabilize the structural OFF state[6], suggesting that myosin head SRX/ADP.$P_i$ conformation might stabilize these interactions.

We tested this idea by measuring the affinity of the myosin motors for their S2 tail domains in the absence and in the presence of F10 and various nucleotide analogues using Microscale Thermophoresis (Fig. S4). Neither change in the nucleotide derivative nor presence of F10 had an effect on the steady-state affinity of the isolated myosin head domain for the S2 tails, suggesting that stabilization of the myosin head SRX state does not increase its affinity for the tail domain in vitro.

Our fluorescence polarization data show that F10 prevents the $Ca^{2+}$-activation dependent re-orientation of the myosin motors towards a more perpendicular orientation, strongly suggesting that it decouples thick filament conformational changes from calcium activation of the thin filament, which was previously also seen for other myosin inhibitors such as Blebbistatin (BS) and Mavacamten[25,28]. Interestingly, BS treatment of ventricular trabeculae strongly reduced the steepness of the force-calcium relation[25]. In contrast, F10 and Mavacamten had only modest effects[50,51], suggesting a different mechanism of force inhibition. BS bound myosin head are likely permanently locked into the OFF state, whereas F10 (and Mavacamten) binding only reduces the likelihood of transition into the strong actin-bound state and F10-bound myosin heads can therefore contribute to myofilament activation via cooperative cross-bridge recruitment. In good agreement, BS has been recently shown to not stabilize the SRX state and induce a different myosin conformation than Mavacamten[52].

Surprisingly, however, in contrast to its inhibiting effect on isometric force and ATPase, F10 increases crossbridge kinetics as indicated by an increase in the rate of force re-development $k_{tr}$ following a slack-stretch protocol (Fig. 4d). A similar effect was observed for Mavacamten using isolated ventricular myofibrils[53]. However, others have reported that Mavacamten slows force development in permeabilized myocardial strips[50,51]. Although the molecular mechanism is currently unknown, it is likely that F10 increases the detachment kinetics of myosin from actin which is not balanced by an equal decrease in the attachment rate (Fig. 7). However, the higher compliance at lower levels of activation might have also affected the rate of force redevelopment. Moreover, cooperative activation of cross-bridges has been proposed to slow the rate of force redevelopment[54]. It follows that F10 might accelerate $k_{tr}$ by stabilizing the myosin head OFF conformation (Fig. 4b), which reduces the intrinsic cooperativity of thick filament activation.

Although no high-resolution structural data are available for Mavacamten bound to the cardiac myosin motor domain, the similar functional and structural effects, as well as the myosin isoform selectivity and chemical similarity of both F10 and Mavacamten suggests that both compounds might bind to the same binding pocket originally identified for Omecamtiv Mecarbil. We have used AutoDock Vina to predict potential binding poses for the F10-derivative R4, Mavacamten and Aficamten in the OM-binding pocket (Fig. S7). All compounds bind deeply into the pocket formed by the converter, N-terminal and lower 50 kDa domain via their terminal Alkyl-groups, and form extensive interactions with both the relay-helix and NTD. Interestingly, a similar docking pose has been predicted for F10 (Fig. 5a), suggesting that these interactions are responsible for the inhibitory effect of cardiac myosin inhibitors. Moreover, the predicted binding affinities of R4, Mavacamten and Aficamten for the OM pocket are significantly higher than that for F10, in good agreement with their lower $EC_{50}$ for inhibiting cardiac myosin function[21,44].

Similar to OM, Mava and Aficamten, F10 shows good selectivity of cardiac over skeletal muscle with no effect on fast skeletal and low to intermediate effect on slow skeletal muscle myofibrillar ATPase[20,21,44]. Although the majority of the residues constituting the OM binding site are highly conserved between the different myosin isoforms, specific residues in converter, L50D and NTD diverge between fast and slow myosin isoforms. Moreover, our docking studies suggest that these residues might be directly involved in interacting with F10's thiourea moiety. In fact, removal of these interactions by truncating the hydrophobic tail completely abolishes the inhibitory effect of F10 (Fig. 6c).

Taken together, our results suggests that the OM-binding pocket can be utilized to modulate myosin motor function to either increase or decrease contractility by small molecule effectors that have different effects on its ATPase cycle. In very good agreement, our limited SAR analysis showed that finely tuned interactions of small molecule effectors occupying the OM-binding site can be utilized to modify myosin function. F10 is a tuneable scaffold for the further development of a novel class of myosin modulators. However, further in vivo testing in animal models are required to establish both the pharmacodynamic and pharmacokinetic profile of F10, followed by optimization of its chemical structure to increase both its bio-availability and potency.

## Methods

### Artificial intelligence-based virtual high throughput screen

The small molecule Virtual High-Throughput Screen (VHTS) was performed as previously described using Atomwise's proprietary AI-based AtomNet® screening platform[55–60]. Briefly, a curated library of small molecules (Mcule_v20201015 containing 4,251,237 compounds), was ranked to identify potential myosin modulators using the Omecamtiv Mercarbil binding site on the human beta-cardiac myosin motor domain (PDB 4PA0)[61] defined by the amino acid residues A91, M92, T94, L96, S118, G119, F121, F489, M493, E497, V698, G701, I702, C705, P710, N711, and R712.

Physicochemical properties were calculated with ICM (Molsoft) and the top ranked 200 molecules were reduced to a list of 84 compounds with drug-like properties (e.g., Lipinski's rule of 5). The 84 selected compounds from the vHTS, along with two negative vehicle controls, were purchased for testing at stock concentrations of 10 mmol $L^{-1}$ in DMSO and validated to be ≥ 85% purity via LC-MS at Mcule. Compounds were blinded during the myosin biochemical assay screening and compound identity and structures were revealed after data were returned to Atomwise. Structural figures were prepared in PyMol v2.4.1.

### Cardiac actomyosin and myofibrillar ATPase measurements

Bovine cardiac myosin S1 was prepared from bovine ventricle as previously described[62]. Rabbit skeletal F-actin was purchased from Cytoskeleton Inc. and prepared for experiments according to manufacturer's instructions.

20 µL of enzyme mix in buffer A (composition in mmol $L^{-1}$: 10 MOPS, 0.1 EGTA, 1 DTT, pH 7) containing 500 nmol $L^{-1}$ bovine cardiac myosin S1, 40 U $mL^{-1}$ lactate dehydrogenase and 400 U $mL^{-1}$ pyruvate kinase were dispensed into a black 96-well half area plate (Greiner). For compound screening 40 nL drug stock (10 mmol $L^{-1}$ in DMSO) were added to each well using a Mosquito liquid handler system. Each assay plate contained five wells with DMSO only (negative control) and five wells with 10 µmol $L^{-1}$ Blebbistatin (positive control). Plates were incubated on a plate shaker at 30 °C for 10 min at 2000 rpm. Reactions were started by adding 20 µL substrate mix in buffer B (composition in mmol $L^{-1}$: 4 MOPS, 9.1 EGTA, 2 $MgCl_2$, 3 $NaN_3$, 1 DTT, pH 7.0) containing 20 µmol $L^{-1}$ F-actin, 440 mmol $L^{-1}$ NADH, 4 mmol $L^{-1}$ 2-phosphoenolpyruvate and 4 mmol $L^{-1}$ ATP using a GILSON PLATEMASTER 96-channel pipette. Plates were briefly mixed by shaking at 5000 rpm on a plate shaker and spun down at 3000 g for 10 s. NADH fluorescence intensity was measured for each well using a ClarioStar Plate Reader for 10 min every 30 sec at 30 °C with the following settings: excitation at 380 nm with a 10 nm bandwidth and emission at 470 nm with a 24 nm bandwidth. Data were recorded with BMG Labtech Reader Control Software v6.20. ATPase activity was extracted by linear regression to changes in fluorescence intensity and normalized to DMSO control using BMG Labtech MARS Analysis Software and GraphPad Prism 10.

Demembranated myofibrils were freshly prepared from bovine ventricle on the day of the experiments as previously described[25,63]. Marmoset ventricular and skeletal muscle tissue was kindly provided by the Biological Service Unit of King's College London. Steady-state myofibrillar ATPase activity was measured using an identical protocol as described above but in myofibril assay buffer (composition in mmol $L^{-1}$: 20 MOPS pH 7, 50 KCl, 0.1 $CaCl_2$, 1 DTT) with a final bovine CMF concentration of 1 mg $mL^{-1}$.

### Single nucleotide turnover experiments

Synthetic thick filaments were freshly prepared by diluting full-length bovine cardiac myosin stock into assay buffer (composition in mmol $L^{-1}$: 15 PIPES pH 7, 5 $MgCl_2$, 1 DTT) and incubation on ice for 2 h. Bovine myosin S1 and synthetic thick filament concentrations were adjusted to 0.4 µmol $L^{-1}$ with assay buffer and 40 µL aliquoted into individual wells of a black 384-well plate (Greiner). Mant-ATP (Jena-Bioscience) was added to each well to a final concentration of 0.4 µmol $L^{-1}$ using an automated injector unit in a ClarioStar Plate Reader (BMG LabTech) and the system was allowed to age for 1 min. The chase phase was started by adding a final concentration of 2 mmol $L^{-1}$ ATP using a second automated injector unit. Fluorescence intensity from each well was constantly measured for 15 min every 1 sec at 25 °C with the following settings: excitation at 360 nm with a 20 nm

bandwidth and emission at 450 nm with a 20 nm bandwidth. For experiments with cardiac myofibrils the concentration was adjusted to 1 mg m L$^{-1}$, and the mant-ATP and ATP concentrations were increased to 20 μmol L$^{-1}$ and 20 mmol L$^{-1}$, respectively.

## Preparation of cardiac trabeculae
All animals were treated in accordance with the guidelines approved by the UK Animal Scientific procedures Act (1986) and European Union Directive 2010/63/EU. All procedures were performed according to Schedule 1 of the UK Animal Scientific Procedure Act, 1986, which do not require ethical approval. All procedures complied with the relevant ethical regulations and were carried out in accordance with the guidelines of the Animal Welfare and Ethical Review Body (AWERB, King's College London).

Wistar rats (*Rattus norvegicus*, male, 200–250 g, 7 weeks old) were killed by cervical dislocation without the use of anesthetics (Schedule 1 procedure in accordance with UK Animal Scientific Procedure Act, 1986) and demembranated right ventricular trabeculae were prepared as described previously[39]. Briefly, free-running, unbranched trabeculae were carefully removed from the right ventricle in oxygenated Krebs-Henseleit solution (composition in mmol L$^{-1}$: 114 NaCl, 5.9 KCl, 1.16 MgSO$_4$, 25 NaHCO$_3$, 0.48 EDTA, 2.2 CaCl$_2$, 5 glucose, pH 7.4) containing 25 mmol L$^{-1}$ 2,3-Butanedione-monoxime (BDM) and demembranated in relaxing buffer (composition in mmol L$^{-1}$: 25 Imidazole, 15 Na$_2$Creatine phosphate (Na$_2$CrP), 78.4 KPropionate (KPr), 5.65 Na$_2$ATP, 6.8 MgCl$_2$, 10 K$_2$EGTA, 1 DTT, pH 7.1) containing 1 % (v/v) Triton X-100.

## Mechanical and fluorescence polarization experiments
Trabeculae were mounted between a strain gauge force transducer (KRONEX, Oakland, California 94602, USA; model A-801, resonance frequency ~2 kHz) and motor (Aurora Scientific, Dublin, D6WY006, Ireland; Model 312 C). BSR-cRLCs were exchanged into demembranated trabeculae by extraction in CDTA-rigor solution (composition in mmol L$^{-1}$: 5 CDTA, 50 KCl, 40 Tris-HCl pH 8.4, 0.1% (v/v) Triton X-100) for 30 min followed by reconstitution with 40 μmol L$^{-1}$ BSR-cRLC in relaxing solution (composition in mmol L$^{-1}$: 25 Imidazole, 15 Na$_2$Creatine phosphate (Na$_2$CrP), 78.4 KPropionate (KPr), 5.65 Na$_2$ATP, 6.8 MgCl$_2$, 10 K$_2$EGTA, 1 DTT, pH 7.1) for 1 h, replacing ~50% of the endogenous cRLC[40,64].

Composition of experimental solutions and activation protocols were identical to those described previously for fluorescence polarization experiments[40]. Fluorescence emission from BSR-cRLCs in trabeculae were collected by a 0.25 N.A. objective using an excitation light beam in line with the emission path. The polarization of the excitation beam was switched at 1 kHz by a Pockels cell (Conoptics) between the parallel and perpendicular directions with respect to the muscle fiber long axis. The fluorescence emission was separated into parallel and perpendicular components by polarizing beam splitters, and its intensity measured by two photomultipliers, allowing determination of the order parameter <*P$_2$*> that describes the dipole orientations in the trabeculae[41]. Force, muscle length and photomultiplier signals were constantly sampled at 10 kHz using dedicated programs written in LabView 2014 (National Instruments). Data were analyzed using Microsoft Excel for Microsoft 365 MSO (Version 2308 Build 16.0) and GraphPad Prism 10.

The sarcomere length of trabeculae was adjusted to 2.1 μm by laser diffraction in relaxing solution prior to each activation. Activating solution contained (in mmol L$^{-1}$): 25 Imidazole, 15 Na$_2$CrP, 58.7 KPr, 5.65 Na$_2$ATP, 6.3 MgCl$_2$, 10 CaCl$_2$, 10 K$_2$EGTA, 1 DTT, pH 7.1. Each activation was preceded by a 2-min incubation in pre-activating solution (composition in mmol L$^{-1}$: 25 Imidazole, 15 Na$_2$CrP, 108.2 KPr, 5.65 Na2ATP, 6.3 MgCl$_2$, 0.2 K$_2$EGTA, 1 DTT, pH 7.1). Solutions with varying concentrations of free [Ca$^{2+}$] were prepared by mixing relaxing and activating solutions using MAXCHELATOR software (maxchelator.stanford.edu). Isometric force and steady-state fluorescence polarization values were measured once steady force had been established. The dependence of force and order parameters on free calcium concentration was fitted to data from individual trabeculae using non-linear least-squares regression to the modified Hill Eq. (1):

$$F = Y_0 + A \cdot ([Ca^{2+}]^{nH})/(-\log_{10}[pCa_{50}]^{nH} + [Ca^{2+}]^{nH}) \qquad (1)$$

where pCa$_{50}$ is the negative logarithm of [Ca$^{2+}$] corresponding to half-maximal change in F, n$_H$ is the Hill coefficient, Y$_0$ is the baseline, and A is the amplitude (for normalized force data: Y$_0$ = 0 and A = 1). Trabeculae which showed a decline in maximal calcium activated force of more than 15% after the experiments were discarded.

## Microscale thermophoresis
Microscale thermophoresis (MST) experiments were performed as described previously[65]. Briefly, MST experiments were performed on a Monolith NT.115 instrument (NanoTemper) in interaction buffer containing 20 mmol L$^{-1}$ Mops, pH 7, 1 mmol L$^{-1}$ MgCl$_2$, 50 mmol L$^{-1}$ KCl, 1 mmol L$^{-1}$ DTT, and 0.05% (v/v) Tween-20. Proteins were labeled with Alexa 647-NHS (Molecular Probes, Inc; Thermo Fisher Scientific) according to the manufacturer's instructions, and dye incorporation (efficiency of >80%) was confirmed by HPLC and ESI–MS. All proteins were either gel-filtered into and/or extensively dialyzed against interaction buffer. Titration experiments were performed with a fixed concentration of 100 nmol L$^{-1}$ of Alexa647-labeled proteins in premium capillaries.

## Langendorff-perfused rat hearts
Rat hearts were Langendorff-perfused as described previously[66]. Briefly, Wistar rats (*Rattus norvegicus*, male, 200–250 g, 7 weeks old) were anesthetized by injection of 0.9 mL sodium pentobarbitone 20% (w/v). Hearts were removed immediately and washed free of blood in ice-cold Krebs-Henseleit solution (composition in mmol L$^{-1}$: 114 NaCl, 5.9 KCl, 1.16 MgSO$_4$, 25 NaHCO$_3$, 0.48 EDTA, 2.2 CaCl$_2$, 5 glucose, 1 Na L-lactate, 0.1 Na-pyruvate, 0.5 L-glutamic acid, 4 hydroxybutyrate, pH 7.4) supplemented with 10 μmol L$^{-1}$ insulin and 0.04% (v/v) intralipid. The aorta was cannulated and secured with sutures (Mersilk 3-0; Ethicon, Somerville, NJ, USA). Hearts were perfused at a constant perfusion pressure of 70 ± 2 mmHg using a peristaltic pump (Gilson Minipuls 4, Middleton, WI, USA) and a feedback control system (STH Pump Controller; AD Instruments, Oxford, UK). Buffer was equilibrated with 95% O2–5% CO2 using a custom-made counter-current membrane oxygenator consisting of spirally wound Silastic tubing (1.47 mm i.d., 1.96 mm o.d.; VWR International, Lutterworth, UK) continually flushed with gas at 37 °C. A fluid-filled balloon, attached to a pressure transducer, was inserted into the left ventricle and inflated to give an end-diastolic pressure between 3 and 8 mmHg. Left ventricular pressure (LVP), perfusion pressure, coronary flow and arterial and venous oxygen tensions were recorded using a PowerLab recorder and LabChart 8.0 software (ADInstruments). Functional parameters were averaged for ~80 cardiac cycles at 5 min intervals.

All animals were treated in accordance with the guidelines approved by the UK Animal Scientific procedures Act (1986) and European Union Directive 2010/63/EU. All procedures were performed according to Schedule 1 of the UK Animal Scientific Procedure Act, 1986, which do not require ethical approval. All procedures complied with the relevant ethical regulations and were carried out in accordance with the guidelines of the Animal Welfare and Ethical Review Body (AWERB, King's College London).

## Statistical analysis
Data are expressed as mean ± standard error (s.e.m.) with the number of independent experiments indicated by n. Normal distribution of data were assessed by Shapiro-Wilks test in GraphPad Prism 8. Statistical significance of differences before and after drug treatment were

assessed with a two-tailed, paired student's t-test or Mann-Whitney test for parametric and non-parametric data, respectively. Non-paired data sets for control and drug group were analyzed using the appropriate non-paired tests. Statistical significance of differences between three or more groups were assessed with a one-way ANOVA followed by Tukey's multiple comparison test.

## Reporting summary

Further information on research design is available in the Nature Portfolio Reporting Summary linked to this article.

## Data availability

The data supporting the findings of the study are available in the article and its Supplementary Information.The structure of Omecamtiv Mecarbil bound to human β-cardiac myosin (PDB code 4PA0) was retrieved from Protein Data Bank. Source data are provided with this paper.

## Code availability

All generated custom written LabView control programs are available from the corresponding author upon request.

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

## Acknowledgements
We are grateful to the British Heart Foundation (BHF) for financial support (PG/19/52/34497 to TK). We like to thank Atomwise Inc. for their support of the study via the AIMS Award program.

## Author contributions
T.K., M.S. and V.K. designed research; P.P., T.K., S.A., Z.H., M.S. and V.K. performed research; P.P., T.K., S.A., Z. H., MS. and V.K. analyzed data; and T.K. wrote the paper.

## Competing interests
The authors declare no competing interests.
