## [Peer Review File · Nature Communications]

REVIEWER COMMENTS

Reviewer #1 (Remarks to the Author):

Parijat and colleagues have used a proprietary AI-based virtual screening tool (Atomwise's AtomNet screening platform) to rank a 4.25M compound library for those that potentially modulate human beta-cardiac myosin near the OM binding site. Biochemical, structural, and physiological secondary assays were used to counter-screen those preliminary hits that experimentally affect cardiac myosin specifically. The identified F10 compound inhibited myosin nucleotide turnover, stabilized the OFF state, reduced force and Ca²⁺-sensitivity. In perfused rat hearts, F10 acted as a negative inotrope. Computational docking was used to suggest where F10 binds near OM's binding site on myosin. F10 may be a new small-molecule compound useful for the development of novel myosin modulators.

This is a clearly articulated and highly interesting manuscript with novel aspects of the approach, findings, and interpretations. Given the importance and usefulness of current myosin modulators as heart disease therapies and research tools, including OM and Mava, the identification through an AI platform and initial MOA characterization of F10 is likely to be of high significance. However, the clarity and impact of this work could be further improved by addressing the following concerns and suggestions.

Results:

1. Errors in Paragraph 1, Sentence 4: distantdistance; -binding sitesites
2. SRX nucleotide release section, last Paragraph: "...slow phase in isolated INTACT bovine cardiac myofibrils..." Aren't these myofibrils skinned or demembranated for the mantATP chase experiments?
3. Should Figure S4 be referred to in the main text Results section rather than first mentioned in Discussion?

Discussion:

1. Has F10 been identified in any other screen or used for any other targets or indications? If there is any existing information about F10 known effects in any systems, this could be compared to the present results with myosin as the target for the HCM indication.
2. 2nd to last paragraph: The authors allude to the idea that "both F10 and Mava...might bind to the same binding pocket originally identified for OM." This is very interesting. Although not the focus of this manuscript, could the authors please provide, in a supplement figure, the docking of Mava to the OM site of myosin? Comparison of the F10 and Mava residue contacts with myosin could then be further

discussed, which could be particularly insightful given the surprising result that F10, despite binding to the OM site, behaves more similarly to Mava.

Methods:

1. Cardiac actomyosin and myofibrillar ATPase measurements (and possibly elsewhere throughout Methods section): When trying to read how “cardiac myofibrils” were prepared, reference #56 (Parijat et al. 2023 Sci. Reports) is provided. However, to find the actual methods of how these were prepared, one must search back through 4 published manuscripts to find any methodological details to Kampourakis et al. 2018 J Gen Physiol. It would be more valuable and efficient for the reader to simply reference this work or reference this work in addition to reference #56. Related to Results Comment #2 above, it would be clearer to state if these preparations are skinned, which they appear to be according to this reference. This type of indirect referencing could be found in other places of the Methods and would be helpful if in place of (or in addition to) the most recent reference, that a reference that includes the protocols is used rather than a reference to a reference (to a reference, to a reference, etc.).

2. Mechanical and fluorescence experiments: Are the years of the programs accurate? For example, was LabView 2014, Excel 2014, and even Prism 9 versions used or was it newer software versions? If applicable, please update to, for example, 2023 or v10 program versions.

Supplement:

1. Figure S2: What V1 and V2 labels of the axes mean are not clear. Can this be further defined in the legend (or axes)? Also, are there units?

2. Figure S4: Should the MST methods be briefly described/referenced in the Methods?

Reviewer #2 (Remarks to the Author):

Parijat and co-authors present a highly innovative and exciting study in this manuscript, demonstrating the potential of in silico AI-based methods to perform meaningful virtual compound screening to identify a biologically active small molecule. Among 84 compounds that emerged from a virtual screen using the binding pocket of a known myosin modulator (omecamtiv mecarbil), one new compound showed a distinctive ability to inhibit cardiac myosin. Other methods for identifying myosin modulators are (presumably) much more costly in time and resources. Beyond the exciting demonstration of this screening approach, the authors present data on the action of the compound that is highly interesting and worthwhile in its own right. Specifically, the new compound (F10) seems to have a much more favorable dosing window and washout speed than the recently FDA approved compound mavacamten.

The manuscript is clear and easy to understand, and the experiments/data analysis appear to be very rigorously done. The most important suggestion I have is that the impact of the work could be increased by acknowledging that there are two valid but different achievements or discoveries to the work: First, this is a striking demonstration of computational drug discovery. Second, the compound identified (F10) has unique characteristics relative to the other known myosin modulators. I believe that the authors should acknowledge this and more fully develop a discussion of their implications separately in the Discussion section. I have shared a few such discussion points below that should be addressed at least in the manuscript text if not via experiments. Although the experiments I suggest are, I believe, very important and compelling, I do not feel they are integral to the publication of the present results.

1. There is no discussion of the fact that there was the narrowest of margins on this screen – exactly one out of 84 was successfully predicted. So the claim that this method could be used to discover more myosin modulators is only reasonable if the 4,251,237 compounds screened represented only a subset of the possible inputs. Is that true? If not, then I believe the authors must acknowledge that they were extraordinarily fortunate. None of the other virtual hits had the faintest glimmer of activity. This approach is also limited in the sense that one must begin with a known binding site such as OM. These limitations scarcely detract from a very exciting accomplishment but should be adequately discussed.

2. On a related note, the discussion claims that this is much more efficient than physical screening of compounds, and indeed it does seem that way, but it would be much better to expand that idea in discussion with some concrete analysis – how much more efficient? Using some standard myosin ATPase screening techniques, how much faster is the virtual screen in terms of cost and time?

3. Is it possible to report how mavacamten, aficamten, and even the R4 variant of F10 would have scored in the virtual screen? It seems like this would be a straightforward test to run, and would provide some additional insight into this work.

4. Is it possible that F10 has a concentration regime in which it can act as a myosin activator, like OM? I suppose it's unlikely, but it is of course well known that above certain concentrations OM inhibits force dramatically. It would be interesting to check low doses of F10 in intact fiber preparations (essentially figure 2b but in intact fibers).

5. I disagree with the interpretation offered in the discussion regarding why F10 would speed ktr. Given the evidence recently produced on multiple fronts of cooperativity within the thick filament for ON-OFF state transitions, the simplest explanation is that F10 stabilizes the OFF state and thereby makes cooperative thick filament recruitment more difficult. Kenneth B. Campbell's seminal analysis on this subject from 1997 ([https://www.cell.com/biophysj/pdf/S0006-3495\(97\)78664-8.pdf](https://www.cell.com/biophysj/pdf/S0006-3495(97)78664-8.pdf)) elegantly argues that ktr is not just a reflection of myosin kinetics but also reveals characteristics of cooperativity. I think this is more likely than an intrinsic effect of F10 to alter myosin detachment kinetics.

6. The extremely rapid wash-out of F10 may be the most exciting discovery presented, given it's enormous potential clinical significance. It would be very striking to present this result in contrast to mavacamten in the same Langendorf system. I believe the authors would do well to consider adding this experiment to enhance the impact of their work.

7. What about medicinal chemistry profile of F10? Is the Langendorf heart experiment already enough to suggest that it could be delivered in vivo? Some comment about what work may remain in developing F10 into a drug would be interesting.

In reading the manuscript, I noticed some minor grammatical/typographical errors:

- Page 3, line 15 – “focus in” is perhaps better as “focusing on”
- Page 4, line 9 – “distant” should be “distance”
- Page 9, fifth line from bottom: “three-time” should be “three-times”
- Page 13, middle: “mammal” should be “mammalian”
- Page 14, top: “a different mechanisms” should be “a different mechanism”
- Page 14: BS abbreviation for blebbistatin is not defined. Also, myosin head should be plural, line 2
- Page 14, middle – omit comma after Although

REVIEWER COMMENTS

Reviewer #1 (Remarks to the Author):

Parijat and colleagues have used a proprietary AI-based virtual screening tool (Atomwise's AtomNet screening platform) to rank a 4.25M compound library for those that potentially modulate human beta-cardiac myosin near the OM binding site. Biochemical, structural, and physiological secondary assays were used to counter-screen those preliminary hits that experimentally affect cardiac myosin specifically. The identified F10 compound inhibited myosin nucleotide turnover, stabilized the OFF state, reduced force and Ca²⁺-sensitivity. In perfused rat hearts, F10 acted as a negative inotrope. Computational docking was used to suggest where F10 binds near OM's binding site on myosin. F10 may be a new small-molecule compound useful for the development of novel myosin modulators.

This is a clearly articulated and highly interesting manuscript with novel aspects of the approach, findings, and interpretations. Given the importance and usefulness of current myosin modulators as heart disease therapies and research tools, including OM and Mava, the identification through an AI platform and initial MOA characterization of F10 is likely to be of high significance. However, the clarity and impact of this work could be further improved by addressing the following concerns and suggestions.

Results:

1. Errors in Paragraph 1, Sentence 4: distantdistance; -binding sitesites

Done. Thank you.

2. SRX nucleotide release section, last Paragraph: "...slow phase in isolated INTACT bovine cardiac myofibrils..." Aren't these myofibrils skinned or demembranated for the mantATP chase experiments?

We agree with the reviewer and have removed the word 'intact'.

3. Should Figure S4 be referred to in the main text Results section rather than first mentioned in Discussion?

We agree with the reviewer and now describe the results of the MST experiments in the results section.

Discussion:

1. Has F10 been identified in any other screen or used for any other targets or indications? If there is

any existing information about F10 known effects in any systems, this could be compared to the present results with myosin as the target for the HCM indication.

*We have performed an exhaustive search of publicly available libraries of bioactive organic small molecules (i.e. ChEMBL and PubChem databases, please see figure below). F10 showed no results in the ChEMBL database using a similarity search with a cut-off of 70% identity. A search in PubChem gave four results, showing that F10 was included as part of a library used for four primary screens against cyclic GMP-AMP synthase, mosquito (*Aedes aegypti*) neuropeptide receptor NPYLR7 and phospholipid biosynthesis in mammalian cells, without, however, any biological activity.*

Figure 1. Screenshots of ChEMBL (left) and PubChem searches (right) for F10.

We have edited the results chapter to emphasize that F10 is a new chemical scaffold without any known biological activity.

2. 2nd to last paragraph: The authors allude to the idea that “both F10 and Mava...might bind to the same binding pocket originally identified for OM.” This is very interesting. Although not the focus of this manuscript, could the authors please provide, in a supplement figure, the docking of Mava to the OM site of myosin? Comparison of the F10 and Mava residue contacts with myosin could then be further discussed, which could be particularly insightful given the surprising result that F10, despite binding to the OM site, behaves more similarly to Mava.

This partly overlaps with point #3 raised by reviewer #2. We have added an additional figure to the Supplementary Information (new Figure S7), showing the top-scoring docking poses of the F10-derivative R4, Mavacamten and Aficamten into the OM binding site. Similar to F10, R4, Mavacamten and Aficamten are predicted to bind deep into the hydrophobic pocket created by the N-terminal domain (NTD), converter and lower 50 kDa domain (L50D). Moreover, all compounds make several contacts with residues in the relay helix of the L50D, which was also observed for F10. In fact, the binding poses of the F10-derivative R4 and Mavacamten are very similar, in good agreement with the EC_{50} of both compounds in the single digit micromolar region.

We have edited the discussion section to further emphasize these points.

Methods:

1. Cardiac actomyosin and myofibrillar ATPase measurements (and possibly elsewhere throughout Methods section): When trying to read how “cardiac myofibrils” were prepared, reference #56 (Parijat et al. 2023 Sci. Reports) is provided. However, to find the actual methods of how these were prepared, one must search back through 4 published manuscripts to find any methodological details to Kampourakis et al. 2018 J Gen Physiol. It would be more valuable and efficient for the reader to simply reference this work or reference this work in addition to reference #56.

Related to Results Comment #2 above, it would be clearer to state if these preparations are skinned, which they appear to be according to this reference. This type of indirect referencing could be found in other places of the Methods and would be helpful if in place of (or in addition to) the most recent reference, that a reference that includes the protocols is used rather than a reference to a reference (to a reference, to a reference, etc.).

We agree, and we have edited the references in the methods chapter accordingly.

2. Mechanical and fluorescence experiments: Are the years of the programs accurate? For example, was LabView 2014, Excel 2014, and even Prism 9 versions used or was it newer software versions? If applicable, please update to, for example, 2023 or v10 program versions.

Thank you for pointing this out. We have updated the version numbers for Excel and GraphPad Prism. However, the control software for the fluorescence polarization experiments was run under LabView 2014.

Supplement:

1. Figure S2: What V1 and V2 labels of the axes mean are not clear. Can this be further defined in the legend (or axes)? Also, are there units?

We have performed the similarity analysis using multi-dimensional scaling in ChemMineTools (Backman et al., 2011, Nucleic Acid Res), which is a form of non-linear dimensionality reduction (similar to principle-component analysis, which uses linear reduction). Therefore, the axis in Figure S2 per se do not have units or descriptors but represent the visualization of multi-variable dataset (e.g. comparing number of atoms, molecular weight, atomic composition, etc) in a two-dimensional plot.

2. Figure S4: Should the MST methods be briefly described/referenced in the Methods?

Thank you for pointing this out. We now included the methods for the MST experiments.

Reviewer #2 (Remarks to the Author):

Parijat and co-authors present a highly innovative and exciting study in this manuscript, demonstrating the potential of in silico AI-based methods to perform meaningful virtual compound screening to identify a biologically active small molecule. Among 84 compounds that emerged from a virtual screen using the binding pocket of a known myosin modulator (omecamtiv mecarbil), one new compound showed a distinctive ability to inhibit cardiac myosin. Other methods for identifying myosin modulators are (presumably) much more costly in time and resources. Beyond the exciting demonstration of this screening approach, the authors present data on the action of the compound that is highly interesting and worthwhile in its own right. Specifically, the new compound (F10) seems to have a much more favorable dosing window and washout speed than the recently FDA approved compound mavacamten. The manuscript is clear and easy to understand, and the experiments/data analysis appear to be very rigorously done. The most important suggestion I have is that the impact of the work could be increased by acknowledging that there are two valid but different achievements or discoveries to the work: First, this is a striking demonstration of computational drug discovery. Second, the compound identified (F10) has unique characteristics relative to the other known myosin modulators. I believe that the authors should acknowledge this and more fully develop a discussion of their implications separately in the Discussion section. I have shared a few such discussion points below that should be addressed at least in the manuscript text if not via experiments. Although the experiments I suggest are, I believe, very important and compelling, I do not feel they are integral to the publication of the present results.

1. There is no discussion of the fact that there was the narrowest of margins on this screen – exactly one out of 84 was successfully predicted. So the claim that this method could be used to discover more myosin modulators is only reasonable if the 4,251,237 compounds screened represented only a subset of the possible inputs. Is that true? If not, then I believe the authors must acknowledge that they were extraordinarily fortunate. None of the other virtual hits had the faintest glimmer of activity. This approach is also limited in the sense that one must begin with a known binding site such as OM. These limitations scarcely detract from a very exciting accomplishment but should be adequately discussed.

The reviewer raises an interesting point. However, there are several lines of evidence that suggest that virtual high throughput screening will likely be very successful in identifying novel myosin modulators.

First, in the current study we only screened a virtual library of about 4.25 million compounds but significantly larger virtual libraries with several billions of compounds are readily available (e.g. ZINC20 and Enamine REAL Space databases; please also see Irwin et al, 2020, J. Chem. Inf. Model), suggesting that potentially hundreds of new chemical scaffolds that act as myosin modulators can be discovered using virtual high throughout screening (vHTS). Moreover, recent estimates suggest that the accessible chemical space for drug-like molecules might be in the region of 10^{26} , which is still only a fraction of total accessible chemical space of $\sim 10^{63}$ (Lu et al., 2022, Journal of Chemoinformatics).

Second, the 84 top-scoring compounds were tested in a biochemical counter-assay monitoring the ATPase activity of isolated myosin S1 in the presence of F-actin. In fact, re-screening the top hits using a phenotypical assay (e.g. cellular contraction assay, which is very feasible given the low compound numbers), might reveal additional hits that modify contractility. For instance,

a molecular binder for myosin S1 might not directly affect the ATPase activity per se but change the intra-molecular interactions that stabilize the myosin head OFF state (i.e. interacting heads motif), which in turn might modulate the number of myosin heads available for contraction.

Third, although we started from a known binding site for OM, computational tools for the prediction of potential binding pockets are readily available and have been used with great success (please see Broomhead et al., 2017, Cell Biochem Biophys).

We have edited the discussion section to further emphasize these points.

2. On a related note, the discussion claims that this is much more efficient than physical screening of compounds, and indeed it does seem that way, but it would be much better to expand that idea in discussion with some concrete analysis – how much more efficient? Using some standard myosin ATPase screening techniques, how much faster is the virtual screen in terms of cost and time?

The reviewer raises another very interesting point. However, this a very complex question that depends on several factors, e.g. the available research infra-structure, personal and equipment, computational power, etc. Moreover, different levels of automation during screening and data analysis will have a significant impact on the time and cost efficiency of the screen. Since those parameters will differ from laboratory to laboratory, a generalized answer cannot be given here.

An example demonstrating the time-efficiency of a virtual vs. wet-lab based high through put primary screen is given below:

The AI-based virtual screen of 4.25 million compounds took about two days (depending on computational power available) + one day for ATPase assay of 86 compounds with four independent repeats + one day for data analysis = 4 days.

*A single post-doc can test 4*96 well plates per working day using the NADH-based assay = 384 compounds per day (excluding control samples). Therefore, screening a standard chemical library with about 100,000 compounds would take about 260 days. Based on previous experience, even a fully automated screening workflow will take about 45 days to screen a library of 4.25m compounds.*

Moreover, for wet-lab based screens all compounds need to be synthesized before the screening, whereas for virtual screen only identified 'hits' have to be synthesized. This reduces cost for synthesis, purification and quality control of required small molecules.

A more detailed analysis of the cost and time efficiency of virtual HTS can be found in Gorgulla, 2023, Annu Rev Biomed Data Sci, and we have added this reference to the discussion section.

3. Is it possible to report how mavacamten, aficamten, and even the R4 variant of F10 would have scored in the virtual screen? It seems like this would be a straightforward test to run, and would provide some additional insight into this work.

This partly overlaps with discussion point #2 raised by reviewer #1. Potential molecular binders in the AI-based virtual screen were ranked using a binary classification model. It follows that the absolute ranks of the top-scoring compounds are less informative as they aren't ranked

on potency, but a binary probability of "being binders," which usually results in small differences in the scores for the top compounds.

However, we have used AutoDock Vina to predict the binding mode and affinity for F10, its derivative R4, Mavacamten and Aficamten for the OM binding site (please see new Figure S7). AutoDock Vina predicted a steady-state dissociation constant K_d of about 28 $\mu\text{mol/L}$ for F10, which is in excellent agreement with the measured EC_{50} of about 20 $\mu\text{mol/L}$ in the myosin S1/F-actin and myofibrillar ATPase assays. As expected from the lower EC_{50} in the myofibrillar ATPase assay, the R4-derivative of F10 shows a predicted higher binding affinity with a K_d of about 17 $\mu\text{mol/L}$. Similarly, both Mavacamten and Aficamten show a two- and four-fold higher affinity for the OM binding pocket than F10, respectively, in good agreement with their higher potency.

We have expanded the discussion section to further emphasize these points.

4. Is it possible that F10 has a concentration regime in which it can act as a myosin activator, like OM? I suppose it's unlikely, but it is of course well known that above certain concentrations OM inhibits force dramatically. It would be interesting to check low doses of F10 in intact fiber preparations (essentially figure 2b but in intact fibers).

We have added an additional figure to the supplementary information (new Figure S5), showing the dose-response relation for the effect of F10 on active isometric tension of demembrated rat trabeculae at about 50% maximal activation. Please note that F10 did not show any activating effect in the concentration range tested (0.625 - 40 $\mu\text{mol L}^{-1}$).

5. I disagree with the interpretation offered in the discussion regarding why F10 would speed ktr. Given the evidence recently produced on multiple fronts of cooperativity within the thick filament for ON-OFF state transitions, the simplest explanation is that F10 stabilizes the OFF state and thereby makes cooperative thick filament recruitment more difficult. Kenneth B. Campbell's seminal analysis on this subject from 1997 ([https://www.cell.com/biophysj/pdf/S0006-3495\(97\)78664-8.pdf](https://www.cell.com/biophysj/pdf/S0006-3495(97)78664-8.pdf)) elegantly argues that ktr is not just a reflection of myosin kinetics but also reveals characteristics of cooperativity. I think this is more likely than an intrinsic effect of F10 to alter myosin detachment kinetics.

We have added this possibility and associated reference to the discussion section.

6. The extremely rapid wash-out of F10 may be the most exciting discovery presented, given its enormous potential clinical significance. It would be very striking to present this result in contrast to mavacamten in the same Langendorff system. I believe the authors would do well to consider adding this experiment to enhance the impact of their work.

We agree with the reviewer, and we have repeated the Langendorff-perfusion experiments in the presence of 1 $\mu\text{mol L}^{-1}$ Mavacamten (new Figure 5, right). The results show that both the ON rate and washout of F10 are several folds faster compared to Mavacamten, which is further quantified in the new Supplementary Figure S6. Please note that we replaced left ventricular developed pressure (LVPD) with left ventricular systolic pressure (LVSP, with

LVSP=LVPD+LVEDP) in Figure 5a, which allowed an easier comparison between the F10 and Mavacamten time-courses.

We have edited the results and discussion chapter accordingly.

7. What about medicinal chemistry profile of F10? Is the Langendorf heart experiment already enough to suggest that it could be delivered in vivo? Some comment about what work may remain in developing F10 into a drug would be interesting.

We have already included the predicted ADMET property for F10 in the previous version of the manuscript (please see Supplementary Figure S3), showing that F10 likely exhibits excellent pharmacodynamics (e.g. high gastro-intestinal absorption), drug-likeness and medicinal chemistry profile. However, we acknowledge that those are only predictions and we have expanded the discussion section to emphasize that in-vivo testing in animal models is required to establish both the pharmacodynamic and pharmacokinetic profile of F10, and optimize its chemistry to increase both its potency and bio-availability.

In reading the manuscript, I noticed some minor grammatical/typographical errors:

- Page 3, line 15 – “focus in” is perhaps better as “focusing on”
- Page 4, line 9 – “distant” should be “distance”
- Page 9, fifth line from bottom: “three-time” should be “three-times”
- Page 13, middle: “mammal” should be “mammalian”
- Page 14, top: “a different mechanisms” should be “a different mechanism”
- Page 14: BS abbreviation for blebbistatin is not defined. Also, myosin head should be plural, line 2
- Page 14, middle – omit comma after Although

Thank you for pointing this out. Done.

REVIEWERS' COMMENTS

Reviewer #1 (Remarks to the Author):

In the revised manuscript, the authors have sufficiently addressed my earlier comments and the comments of the other reviewer, including expansion of the discussion and new data.

Reviewer #2 (Remarks to the Author):

The authors have been very attentive to the comments offered in the original reviews. In particular, they have opted to add critical experiments that bolster their claims and highlight properties of the newly identified compound (F10) that differ from other known myosin inhibitors.